# Multidimensional Feature in Emotion Recognition Based on Multi-Channel EEG Signals

**DOI:** 10.3390/e24121830

**Published:** 2022-12-15

**Authors:** Qi Li, Yunqing Liu, Quanyang Liu, Qiong Zhang, Fei Yan, Yimin Ma, Xinyu Zhang

**Affiliations:** 1Department of Electronics and Information Engineering, Changchun University of Science and Technology, Changchun 130000, China; 2Economics School, Jilin University, Changchun 130000, China

**Keywords:** EEG, emotion recognition, multidimensional feature, depthwise separable convolution

## Abstract

As a major daily task for the popularization of artificial intelligence technology, more and more attention has been paid to the scientific research of mental state electroencephalogram (EEG) in recent years. To retain the spatial information of EEG signals and fully mine the EEG timing-related information, this paper proposes a novel EEG emotion recognition method. First, to obtain the frequency, spatial, and temporal information of multichannel EEG signals more comprehensively, we choose the multidimensional feature structure as the input of the artificial neural network. Then, a neural network model based on depthwise separable convolution is proposed, extracting the input structure’s frequency and spatial features. The network can effectively reduce the computational parameters. Finally, we modeled using the ordered neuronal long short-term memory (ON-LSTM) network, which can automatically learn hierarchical information to extract deep emotional features hidden in EEG time series. The experimental results show that the proposed model can reasonably learn the correlation and temporal dimension information content between EEG multi-channel and improve emotion classification performance. We performed the experimental validation of this paper in two publicly available EEG emotional datasets. In the experiments on the DEAP dataset (a dataset for emotion analysis using EEG, physiological, and video signals), the mean accuracy of emotion recognition for arousal and valence is 95.02% and 94.61%, respectively. In the experiments on the SEED dataset (a dataset collection for various purposes using EEG signals), the average accuracy of emotion recognition is 95.49%.

## 1. Introduction

Emotion recognition is a multi-curricular field of study that combines cognitive science, social psychology, and electronic information science [1,2,3]. It is a complex problem and a hot spot in the cognitive science industry. Emotion identification methods are divided into two categories. One is emotion identification based on non-physiological data signals, such as words, language expressions, facial emotions, gestures, etc. [4]. However, this method is relatively subjective and cannot ensure the authenticity and effectiveness of emotions. Another emotion recognition is based on physiological data signals [5]. With the development of wearable and non-invasive physiological signal acquisition devices, the real-time performance and accuracy of the physiological signal collection have been greatly improved, promoting the development of physiological signals in emotion recognition. Nervous system physiology and social psychology studies have shown that brain electrical signals can reflect various brain wave thematic activities and effects of the brain. Since the occurrence of emotion is related to the activities of the cerebral cortex, the brain’s electrical signals can also reflect the efficient information content of people’s emotional state [6]. Because of the advantages of strong universality and high classification and identification accuracy, brain electronic signals have been gradually introduced into the field of emotion identification research.

Traditional EEG emotion identification methods usually capture handcrafted emotion characteristics, followed by shallow machine learning for emotion classification. The main methods include the K-nearest neighbor algorithm, mixed Gaussian algorithm, artificial neural network method, Bayesian algorithm, support vector machine, and improvements in these methods [7]. At this stage, the key characteristics of emotional discrimination of EEG signals are time-domain characteristics, frequency domain characteristics, and time frequency domain characteristics [8]. Time domain features are mainly used to directly extract the waveform features of EEG signals, including event related potential (ERP), signal statistics, energy, power, high order zero crossing analysis, non-stationary index (NSI), and fractal dimension (FD) [9]. Since the time-domain feature cannot display the frequency information of the signal, the researchers added frequency domain analysis. First, the original time domain signal is converted into the frequency domain to obtain the frequency spectrum. Then, the frequency band is decomposed into five sub-frequency bands (*δ*, *θ*, *α*, *β*, *γ*) that are closely related to human psychological activities [10]. Extract features from these five frequency bands. These features usually include power spectral density (PSD) [11], differential entropy (DE) [12,13], differential asymmetry (DASM) [14], and rational asymmetry (RASM) [15,16]. EEG signal is a non-stationary and non-linear random signal [17]. Since the scope of the Fourier transform is the entire time domain, it lacks local generalization ability. It is impossible to confirm the time corresponding to each frequency domain component of the non-stationary signal, so the time frequency domain is introduced to analyze the EEG signal [18]. Usually, the time frequency domain signal transformation mainly adopts methods, such as short time Fourier transform (STFT), wavelet transform (WT), and wavelet packet transform (WPT) or Hilbert–Huang [19]. However, these features need to be designed and selected according to specific purposes, which also limits their performance in EEG emotion classification.

The rapid development of deep learning technology in computer vision [20], speech [21], and natural language processing [22], methods based on deep neural networks, have begun to get the attention of researchers in this field. EEG usually contains much helpful information and has a wide range of applications. The related work is summarized in Table 1. Komolovaitė et al. [23] contributed to the application of Alzheimer’s disease. They used convolutional neural networks to learn discriminant features in EEG to classify familiarity and emotion. Thammasan used deep belief networks combined with fractal dimension, power spectrum, and discrete wavelet transform features to classify the emotional categories of EEG under music stimulation [24]. Tripathi gives the effect of the convolutional neural network in the two classifications (high/low) and three classifications (high/normal/low) of EEG emotions [25]. Elham scientifically investigated a three-dimensional convolutional neural network (3D-CNN) for emotion discrimination based on multichannel EEG data information, and the discrimination accuracy of valence and arousal were 87.44% and 88.49%, respectively [26]. Yang obtained the differential entropy characteristics of different frequency bands and applied the three-dimensional convolutional neural network to classify emotions on DEAP data at the level of valence and arousal. The mean accuracy is 89.45% and 90.24%, respectively [27].

Although the deep learning emotion recognition model has achieved higher accuracy than the shallow model, how to integrate more helpful EEG information to perform emotion recognition better and improve the validity and reliability of predictions made with machine learning remain challenging [28]. Firstly, regarding the relationship between EEG signal frequency bands and emotion types, Zheng et al. [29] applied a deep belief network model to select meaningful frequency bands of key channels by training the weight distribution of DBNs. The results showed that the Gamma and Beta bands exhibited better classification performance in emotion recognition. Secondly, EEG signal acquisition equipment generally includes 32 channels, 62 channels, and 128 channels. It may contain some EEG signals that have nothing to do with emotions, which can bring noise and interference to emotion recognition. In response to this problem, Meng-meng extracted the frequency domain features on each electrode and then combined the common spatial patterns (CSP) algorithm with the spatial domain to calculate the spatial frequency domain features. The average accuracy rate in the three-category emotion recognition has reached 87.54% [30]. Finally, combining the temporal and spatial characteristics of EEG signals, Liu designed the RCNN LSTM hybrid deep neural network entity model, using the circulatory system neuron network to obtain the characteristics of brain electrical signals, then integrating the information content of the LSTM time concept characteristics to develop a fusion model [31].

In summary, the frequency, time, and space characteristics of EEG signals are critical for emotion identification. However, most of the literature only considers one or two of these three functions. Therefore, it is possible to adaptively capture important frequency, spatial and temporal characteristics in a unified network by integrating information from different fields, which is of crucial significance for improving the accuracy of EEG signal emotion recognition. The method proposed in this paper makes full use of frequency, temporal, and spatial information features while also making an outstanding contribution to the computational performance of the network, filling a gap in integrating information while improving computational performance in EEG signal emotion recognition.

A single feature extracted by traditional emotion research methods is not enough to describe the rich information contained in EEG, and a simple feature combination may cause redundant features in the feature space, thereby affecting model accuracy and increasing model complexity. This paper points out a new method of EEG emotion discrimination based on multi-domain feature fusion. The general flowchart of the proposed method is shown in Figure 1. Firstly, the EEG signal is converted into a 4D feature structure composed of several time slices, which contain frequency, spatial and temporal information. Secondly, because the EEG data training samples are limited and the feature dimension is large, we use the depthwise separable convolution for dimensionality reduction, which balances the width and depth of the network and can reduce the amount of computation. The frequency and spatial features of EEG signals are fully extracted by using the feature of depthwise separable convolution. Finally, since the traditional LSTM does not consider the order information of updating neurons during feature learning, it means that each neuron is independent. Therefore, the knowledge learned from the perspective of spatial structure is discrete, which makes it difficult to distinguish the information level between neurons, which causes the loss of information at the spatial level. Therefore, this paper uses ON-LSTM to increase the induced bias towards structural information by sorting neurons, which enables the model to integrate the emotional hierarchy of EEG signals into LSTM through a specific neuron order without destroying its sequential form. Thereby, the feature information of the emotional hierarchy of the EEG signal is learned to express the hidden emotional information in the time series.

The key dedications in the text are as follows:We adopt a four-dimensional (4D) feature structure, including the frequency, space, and time information of EEG signals, as the feature input of EEG signal emotion recognition.We use the FSTception model based on a depthwise separable convolutional neural network to solve the problems of a few training samples of EEG data, large feature dimensions, and feature extraction.In particular, we adopt the ON-LSTM structure to deal with the deep emotional feature extraction hidden on time series in input features with a 4D structure.

Other parts of the organization are listed below. We will introduce the method proposed in this article in Section 2. Section 3 introduces the data set, experimental settings, results, and discussion. Finally, in Section 4, we conclude that we have achieved advanced performance on both the SEED and DEAP data sets.

## 2. Materials and Methods

This section mainly describes the 4-dimensional feature organization, FSTception model, and classification. The small details of each part will be introduced in detail.

### 2.1. 4D Frequency Spatial Temporal Representation

The original EEG signal can be divided into several different frequency patterns according to the in-band correlation of different behavioral states. The frequency modes and corresponding characteristics of EEG are listed in Table 2. The awareness increases with the increase of the frequency band. We believe that emotions are generated when humans are highly awake. This statement also conforms to our intuitive understanding. If a person is unconscious, he/she is unlikely to have specific emotions. To extract the feature entirely, we constructed a 4D feature structure to integrate the three information features of EEG signals, as shown in Figure 2. In data processing, just like the previous work [26,32], because the Butterworth filter has the smoothest response curve in the passband and is simple in design and easy to implement, we use the Butterworth filter to divide the EEG signal into five frequency bands (*δ*[1~4 Hz], *θ*[4–8 Hz], *α*[8~13 Hz], *β*[13~30 Hz], *γ*[30~50 Hz]). The frequency response of the filter is shown in Figure 3. After Butterworth filtering decomposition, the EEG data of a participant varied from 40 × 8064 × 32 (video × sample × channel) to 40 × 8064 × 5 × 32 (video × sample × band × channel). The EEG raw signal and its five frequency bands in the DEAP dataset are shown in Figure 4.

Since our focus is on identifying emotional states at the segmented level, the EEG signal is cut into l segments of degree n. Our goal is to categorize these fragments into the correct tags. The segmented EEG data is converted into 40 × l × n × 5 × 32 (video × segment × length × band × channel). When 0.5s is taken as the cutting window, l=120 is the number of segments to be cut, and n=64 is the sample points contained in a segment. Differential entropy can accurately reflect the response degree of the stimulus because of its ability to distinguish the balance between the low frequency band and the high frequency band of the EEG mode [33]. Therefore, the differential entropy algorithm is used in this experiment to represent the complexity of a fixed length EEG signal. Differential entropy is the generalized form of Shannon’s information entropy on continuous variables, where X is a random variable, and its original expression is
(1)H(X)=−∫f(x)log[f(x)]dx
where *f*(*x*) is the probability density function of the X. Shi et al. [34] proved that the differential entropy of a random variable approximately obeying the Gaussian distribution *N*(*θ*, *σ*^2^) is equal to the logarithm of its energy spectrum in a specific frequency band. The specific expression is as follows
(2)$$H\left(X\right)=-\int_{-\infty }^{+\infty }\frac{1}{\sqrt{2\pi \sigma^{2}}}e^{-\frac{{\left(x-\mu \right)}^{2}}{2\sigma^{2}}}\log\left[\frac{1}{\sqrt{2\pi \sigma^{2}}}e^{-\frac{{\left(x-\mu \right)}^{2}}{2\sigma^{2}}}\right]dx=\frac{1}{2}\log2\pi e\sigma^{2}$$

The original EEG signal is complex and does not obey the normal distribution, but the bandpass filtering of a specific range approximates the Gaussian distribution. In order to test whether the EEG after band-pass filtering follows the Gaussian distribution, 4800 EEG segments of 0.5 s length in each sub-band signal were randomly selected from 32 subjects. The Kolmogorov–Smirnov test was then applied to all 4800 EEG data segments to test whether each sub-band was subject to Gaussian distribution, and the significance level α was set at 0.05. It can be proven that the probability of a sub-band signal satisfying the Gaussian distribution hypothesis is greater than 90%.

Therefore, in the fixed frequency band i, the DE feature of each EEG segment in each frequency band is respectively calculated as
(3)$$H_{i}\left(X\right)=\frac{1}{2}\log\left(2\pi e\sigma_{i}^{2}\right)$$
where $H_{i}$ and $\sigma_{i}^{2}$ represent the differential entropy and signal variance of corresponding EEG signals in the frequency band i, respectively.

From Equation (3), where π,e are constants, we only need to know $\sigma_{i}^{2}$ to get the differential entropy of X. For the frequency band of the EEG signal, as a result of the zero mean, the variance can be estimated as
(4)$${\hat{\sigma }}^{2}=\frac{1}{N}\sum_{i=1}^{N}x_{i}^{2}$$

Equation (4) shows that the variance estimate of the signal sequence *X* is its average energy. Moreover, due to Parseval’s theorem, the average energy is related to the energy spectrum. It can be written as Equation (5) For the Discrete Fourier transform
(5)$$\sum_{n=0}^{N-1}\left|x_{i}^{2}\right|=\frac{1}{N}\sum_{k=0}^{N-1}\left|X_{k}^{2}\right|=P_{i}$$
where $X_{k}$ is the discrete Fourier transform coefficient of signal sequence $X_{i}$. N is the length of the fixed time windows. $P_{i}$ can be thought of as the energy spectrum, equal to the value of the signal variance $\sigma_{i}^{2}$ multiplied by a constant coefficient N.

Therefore, after the bandpass filter to a specific band *i*, its variance can be estimated by $\frac{P_{i}}{N}$. The relationship between the logarithmic energy spectrum and differential entropy can be expressed as
(6)$$\begin{array}{l} H_{i}\left(x\right)=\frac{1}{2}\log\left(2\pi e\sigma_{i}^{2}\right)=\frac{1}{2}\log(N\sigma_{i}^{2})+\frac{1}{2}\log(\frac{2\pi e}{N})\\ \;\;\;=\frac{1}{2}\log\left(P_{i}\right)+\frac{1}{2}\log\left(\frac{2\pi e}{N}\right) \end{array}$$

It can be seen that for EEG series of fixed length, the estimate of differential entropy is equivalent to the logarithmic energy spectrum of a specific frequency band.

EEG signals have characteristic frequency ranges and spatial distribution, which are often associated with different functional states of the brain. In the electroencephalogram, each electrode is physically adjacent to the other and records the electrical signals in one area of the brain. In order to maintain spatial information between multiple adjacent channels, according to the mapping method shown in Figure 5, the one-dimensional EEG signal is constructed into a two-dimensional matrix with a size of h×w. The values of h and w are the maximum values of the electrodes in the vertical and horizontal directions, respectively. Among them, a value of zero indicates that the signal of the channel is not used. Then, the two-dimensional feature matrices of different frequency bands are superimposed at the same time to obtain a three-dimensional matrix h×w×a, where a indicates the total number of frequency bands. The 4D structure can be expressed as $X_{n}\in R^{h\times w\times a\times 2T}$, n=1,2,⋯,S, where S is the total number of samples. In this work, the values of h and w were determined according to the two-dimensional matrix mapping of the distribution positions of EEG acquisition electrodes. Two methods were used to select a compact graph (8 × 9) and a sparse graph (19 × 19). 

### 2.2. The Structure of FSTception

The structure of the FSTception model proposed in this paper consists of two parts, the first is that we use a deep separable convolutional network to further extract the frequency and spatial information of the 4D input feature structure. Secondly, the LSTM with ordered neuron structure is used to extract the contextual correlation of the time-series EEG signals, thus mining the deep emotional information of the EEG signals and improving the accuracy of emotion recognition.

#### 2.2.1. Frequency Spatial Characteristic Learning

Both spatial and temporal information is vital for time-series EEG signals, which represent activity relationships at different locations in the brain and the dynamics of brain patterns over time. In order to better represent the spatial features of EEG information in limited training samples, we propose the FSTception model, a CNN structure based on depthwise separable convolutions. For the 4D frequency space time structure expressed as $X_{n}$, we apply the FSTception model to obtain frequency and spatial information from each time slice. The structure is shown in Figure 6. First, since the size of the 2D feature maps in the input samples is 8 × 9, the two-dimensional map size is relatively small. In order to better preserve all the information, we set up two convolutional layers, the first convolutional layer has 64 feature maps, and the filter size is 1 × 1, the second convolutional layer has 128 feature maps, the filter size is 3 × 3, and the deep features are extracted using convolution kernels of different scales. Then, we use deep convolution to learn spatial features. The main benefit of deep convolution is to reduce the total number of parameters in the network training process because the convolution is not fully connected to the previous feature map but are connected to each feature map separately to learn spatial features of specific frequencies. Importantly, when used in specific EEG applications, this operation provides a direct method to learn the spatial characteristics of each time slice so that the spatial characteristics of specific frequencies can be effectively extracted. Finally, we used separable convolution, which is a combination of deep convolution and pointwise convolution. The main advantages are: (1) reduce the number of fitting parameters; (2) summarize its feature maps separately by learning each core, and then merge the optimal results and output, thereby expressing the relationship between the internal and cross-feature maps of the feature map. When performing emotion recognition on EEG, this operation is used to summarize a single feature map (depthwise convolution) and optimize the combined feature map (pointwise convolution), because different feature maps can represent information at different time scales. For the convolutional layers of this network, zero-filling and rectified linear unit (ReLU) activation functions and normalization operations are used. To better reduce overfitting and improve the robustness of the network, we employ a max pooling layer with a size of 2 × 2 and a step size of 2 after the operation of the convolution neural network. Finally, the output of the max-pooling layer is unrolled and sent to a fully connected layer with 512 units. The frequency and spatial features of the final output EEG are expressed as $Q_{n}=R^{512\times 2T}$.

#### 2.2.2. Temporal Characteristic Learning

Since the EEG signal contains dynamic content, the changes between time slices in the 4-dimensional structure may hide other information, which may help perform more accurate sentiment classification. Considering that the long short-term memory (LSTM) network has better modeling capabilities for time series information, this paper uses ON-LSTM to integrate neurons into the LSTM through specific sorting, thereby allowing ON-LSTM to learn the hierarchical structure information automatically [35]. The detailed structure of ON-LSTM is shown in Figure 7. ON-LSTM is actually a change in the overall design of the LSTM cell state update. The cell state is done in a hierarchical operation, and the traditional LSTM is to update all the cell states. The cell state is out of order, while ON-LSTM to cell state is updated by two integers for the hierarchy and then hierarchical for computation. This paper is the first innovative use of ON-LSTM in the temporal analysis of EEG emotional features. ON-LSTM is an improved network structure based on LSTM, given a CNN output encoding sequence *Q_n_* = (*q_1_*, *q_2_*, *q_2T_*), where *q_t_* = *R*^512^ and *t* = 1, 2, 2*T*. We apply an ON-LSTM layer with 128 data stores to discover the temporal dependence of internal structural segments. The gate structure and output structure of ON-LSTM are the same as those of general LSTM:(7)$$f_{t}=\sigma \left(W_{f}q_{t}+U_{f}h_{t-1}+b_{f}\right)$$
(8)$$i_{t}=\sigma \left(W_{i}q_{t}+U_{i}h_{t-1}+b_{i}\right)$$
(9)$$o_{t}=\sigma \left(W_{o}q_{t}+U_{o}h_{t-1}+b_{o}\right)$$
(10)$${\hat{c}}_{t}=\tan h\left(W_{c}q_{t}+U_{c}h_{t-1}+b_{c}\right)$$
(11)$$h_{t}=o_{t}{}^{\circ}\tan h\left(c_{t}\right)$$
where historical information $h_{t-1}$ and current information are input, the input gate $i_{t}$ determines the amount of change of the input vector $q_{t}$ to the information in the storage unit at the current moment. Output gates $o_{t}$ manipulate the output of information contained in the current storage unit. The forget gate $f_{t}$ determines the degree of influence of the historical information $h_{t-1}$ at the last moment on the information in the current storage. $h_{t}$ is the final output of the ON-LSTM unit at a time t. σ(⋅) is the logistic sigmoid function, tanh(⋅) is the hyperbolic tangent function, $W_{\cdot }$ and $U_{\cdot }$ are the weight matrix corresponding to each gate. $b_{\cdot }$ the bias term corresponding to each gate. ° is the corresponding element multiplication.

The improvement of ON-LSTM relative to LSTM is different in the update mechanism from ${\hat{c}}_{t}$ to $c_{t}$, thus introducing a hierarchical structure of based on LSTM.
(12)$${\tilde{f}}_{t}=\vec{cs}\left(softmax\left(W_{j}q_{t}+U_{j}h_{t-1}+b_{f}\right)\right)$$
(13)$${\bar{i}}_{t}=\overset{\leftarrow }{cs}\left(softmax\left(W_{i}q_{t}+U_{i}h_{t-1}+b_{i}\right)\right)$$
(14)$$w_{t}={\tilde{f}}_{t}{}^{\circ}{\tilde{i}}_{t}$$
(15)$$c_{t}=w_{t}{}^{\circ}\left(f_{t}{}^{\circ}c_{t-1}+{\tilde{i}}_{t}{}^{\circ}{\tilde{c}}_{t}\right)+\left({\tilde{f}}_{t}-w_{t}\right){}^{\circ}c_{t-1}+\left({\tilde{i}}_{t}-w_{t}\right){}^{\circ}{\tilde{c}}_{t}$$
where $\vec{cs}$ is cumsum from left to right, and $\overset{\leftarrow }{cs}$ is cumsum from right to left. The main forget gate ${\tilde{f}}_{t}$ determines the level of influence to the hierarchical data in the current data storage from the hierarchical information content included in the historical in-formation *h*_*t*−1_ at the last moment. The main input gate ${\tilde{i}}_{t}$ determines the amount of change of the level information contained in the input vector *q_t_* at the current moment to the level information in the storage unit. After ON-LSTM sorts the neurons, the information level is represented by the front and back of the position. Then, when updating the neuron, first predict the historical level *h*_*t*−1_ and the input level *q_t_*, respectively, and update the neurons between partitions through these two levels. ON-LSTM partition update is shown in Figure 8. In this way, the high-level information may retain a considerable distance because the high-level will directly copy the historical information so that the historical information is continuously copied without being changed, and the low-level information may be updated at each step of input. Because the lower layer directly copies the input, and the input is constantly changing, a hierarchical structure is formed through information classification. More generally speaking, it is a group update. Higher group information is transmitted farther (larger span), and lower group span is more minor. These different spans form the hierarchical structure of the input sequence. By learning the hierarchical structure information, it can better mine the hidden connections between each time slice and then achieve more accurate sentiment classification.

The final node output of ON-LSTM is used to represent the highest level of the EEG segment, *Y_n_* ∈ *R*^128^, which integrates the frequency, time, and space features of the EEG segment.

### 2.3. Classification

According to the final characteristics representation *Y_n_*, we apply the fully connected layer and the softmax activation function to predict and analyze the sample label of the EEG segment *X_n_*. The calculation is as follows
(16)$$Out=AY_{n}+b=\left[out_{1},out_{2},\cdots ,out_{c}\right]$$
where *A* represents the transform matrix, *b* represents the bias, and *m* represents the number of emotion categories. Then, according to the output result, it is sent to the softmax classifier to obtain the final emotion discrimination result., which can be described as
(17)$$P\left(c|X_{n}\right)=\frac{\exp\left(out_{j}\right)}{\sum_{i=1}^{i=c}\exp\left(out_{i}\right)}\left(j=1,\cdots ,c\right)$$
where represents the normalized prediction results of each category. In back propagation, the Adam gradient algorithm is used.

## 3. Results

### 3.1. Experimental Setup

The tasks in this paper are done on TensorFlow on a suitable GPU (Nvidia Quadro P5000) using Python 3.7 and Keras. For each trial, we completed a five-fold cross validation and measured the mean classification accuracy (ACC) and standard deviation (STD) for each experimenter. The final characteristics of everyone’s mode are shown as the mean ACC and STD of all subjects. Applying the Adam boost loss function, we set the learning rate to 0.003, the batch size to 120, and the epoch to 100. Codes are available at https://github.com/lixiaonian2022/code.git (accessed on 14 July 2022).

All EEG trials are divided into five groups randomly, then four folds of trials are combined as training data, and the left 1-fold is considered as test data. The experiments are repeated five times so that every fold has a chance to be the test data. The result is calculated as the average of five experiments.

### 3.2. Dataset

All subjects were in good physical and psychological health, and 32 conductor AgCl electrodes were used in the collection of data, with scalp electrodes positioned in accordance with the international standard 10–20 system.

#### 3.2.1. SEED Dataset

The SEED data set [36] has a total of 15 physical and mental health subjects (seven men, eight women, mean age 23.27). Each subject performed three sets of experiments, and each experiment watched 15 movie clips (five positive segments, five negative segments, five neutral segments). The viewing process of each segment can be divided into four stages, including a 5-s start prompt, 4-min movie playback time, 45-s self-assessment, and 15-s rest time. During the film screening, two films with the same emotion will not be displayed consecutively. Each volunteer carried out three experiments, each time separated by one week, for a total of 45 experiments.

#### 3.2.2. DEAP Dataset

The DEAP data set [37] contains 32 subjects, and each subject has 32 channels of EEG signals and 8 channels of peripheral physiological signals. The content of the DEAP dataset is shown in Table 3. The 32 channels EEG signal is used as the experimental data in this paper. The EEG data signal was first sampled at a sampling frequency of 512 Hz. Then, the sampling frequency was reduced to 128 Hz, and then filtered using a 4.0~45.0 Hz filter to remove EOG artifacts. Each subject watched 40 one-minute emotional music videos. After watching each short video, the subjects scored the level of arousal, valence, liking, and controlling influence. In this experiment, only two dimensions of arousal and potency were selected for testing.

### 3.3. EEG Data Preprocessing

In order to eliminate the influence of artificial interference, such as ECG, EMG, and power frequency interference, on emotion recognition, we retained more practical components of emotional EEG. The raw multichannel EEG data were first filtered using a Butterworth bandpass filter, using 1 Hz and 50 Hz limits to divide the frequency bands into five bands containing emotional information. These include the δ band (1 to 4 Hz), the θ band (4–8 Hz), the α band (8–13 Hz), the β band (13–30 Hz), and the γ band (30–50 Hz). Each trial contains an EEG signal of 63 s. Among them, the first 3 s is the baseline signal, and the 60 s is the emotional signal of the EEG.

The EEG of 63 s was divided into 126 segments with a length of 0.5 s on average, the first 3 s was the baseline signal, and the eigenvalue of each frequency band of each segment was calculated. In the experiment, in order to improve the accuracy of the experiment, the difference between the characteristic value of EEG recorded by the subject under the stimulus and the mean characteristic value of the baseline signal was used as the experimental data, and the difference at the moment was recorded as the average segmentation of the 63 s EEG into 126 segments with the length of 0.5 s, in which the first 3 s was the baseline signal. The eigenvalue of each frequency band of each segment is calculated. In the experiment, in order to improve the accuracy of the experiment, the difference between the characteristic value of EEG recorded by the subject under the stimulus and the mean characteristic value of the baseline signal was used as the experimental data, and the difference at time t was denoted as $EEG\_removed_{t}$. The calculation formula is as follows
(18)$$\begin{array}{l} EEG\_removed_{t}=\\ feature_{t}-\frac{base\_feature_{1}+base\_feature_{2}+\cdots +base\_feature_{6}}{6} \end{array}$$
where $feature_{t}$ is the EEG characteristic value extracted at time t and $base\_feature_{i}(i=1,2,\cdots ,6)$ is the baseline signal characteristic value extracted at time i.

### 3.4. Results and Discussion

The research results of the 4D-FSTception method proposed in this article on the SEED data set is shown in Table 4, and as shown in bar Figure 9, numerical changes can be observed and compared more easily. The accuracy rate of all subjects exceeded 90%, and the average accuracy rate reached 95.49%. For different subjects, the classification accuracy rate is different. For subject 1, the average classification accuracy rate in the three time periods is 91.5% (the lowest), and for subject 4, the classification accuracy rate in the three time periods is 91.5%. The average classification accuracy rate is 99.59% (the highest), which reflects individual differences. However, for the same subject, the classification accuracy of different time periods fluctuates. For example, for subject 8, the classification accuracy rate in time period 2 is 94.81% (lowest), and the classification accuracy rate in time period 3 is 97.63% (highest). Among 15 subjects, eight items (#2, #4, #6, #8, #9, #10, #11, #13) are better than the mean classification accuracy.

The experimental results of the 4D-FSTception method on the DEAP dataset are shown in Table 5 and Table 6, respectively, and as shown in bar Figure 10. The average accuracy of arousal classification is 95.02%, and the standard deviation is 1.85%. Only two subjects (#2 and #22) had an accuracy rate less than 90%. In particular, the valence and arousal accuracy rates of tester #22 were 79.88% and 84%, respectively, which were lower than other testers. The reason for this may be that the subjects were inattentive during the test operation, or the level of subjective experience was not effectively reported after the test.

The accuracy, precision, recall, and F1-score metrics are used to evaluate the performance of the 4D-FSTception. The evaluation index is calculated based on all the predicted results in DEAP and SEED datasets, as shown in Table 7 and Table 8. The confusion matrix of different data sets is shown in Figure 11 and Figure 12. The superiority of the model in this paper in EEG signal emotion recognition was fully demonstrated.

In order to verify the computational performance of the methods in this paper, we will choose different methods for time cost analysis in both SEED and DEAP datasets. To be able to view and compare maximum and minimum values in a more straightforward way. Figure 13 and Figure 14, respectively, show the performance of accuracy and time cost by different methods in different datasets. As shown in Table 9, for the SEED dataset, a comparison between the HCNN method with a sparse input map of 19 × 19 and the 4D-CRNN method with the same input map of 8 × 9 as in this paper was used. After training, it can be seen that the accuracy of the compact graph is 95.49%, which is 6.89% higher than that of the sparse graph. Furthermore, in terms of their time cost, the compact map spends only 660 s per topic in the five-fold cross-validation, which is almost a quarter of the time cost of the sparse map. Moreover, it can be found that the accuracy of our compact map is close to that of the 4D-CRNN method, which is also a compact map. However, in terms of time cost, the method in this paper is 143 s faster than the 4D-CRNN method. On the DEAP dataset, our compact map outperforms the sparse map and outperforms the 4D-CRNN method with the same compact map in terms of accuracy and training time cost. For valence classification, the average accuracy of our method is 95.49%, which is 0.42% higher than that of the 4D-CRNN method with the same input map. In addition, the time cost of this paper’s method is even 15 s faster than that of the 4D-CRNN method. For arousal classification, we can obtain a similar conclusion. The reason for obtaining similar results may be that adding a sparse map of zeros between adjacent electrodes does not add any useful information. The lower time cost of the compact map may be due to its smaller size, which involves fewer convolution filters to compute. We obtained the model parameter quantities as well as the floating-point operations to be 12.7 M and 20.61 G, respectively, in the DEAP dataset. Compared with other methods, the algorithm in this paper has better performance. The results for both datasets lead us to believe that the method in this paper exhibits better computational performance when classifying emotions.

To verify the importance of the combined structure of our proposed CNN model and ON-LSTM, we performed two different combinations of CNN + LSTM (4D-FSTception (LSTM)) and CNN + ON-LSTM (4D-FSTception) in the SEED and DEAP datasets. The comparative experimental results are shown in Figure 15. Structure experiment. The average classification accuracy of 4D-FSTception in the SEED dataset and the DEAP dataset is 0.4% higher than the average classification accuracy of 4D-FSTception (LSTM). The results show that the ON-LSTM structure contributes to modeling time-series information in EEG signal emotion recognition. The hidden connection between each time slice can be better mined by learning the hierarchical structure information. In particular, the output results of the network proposed in this paper, after the EEG frequency and spatial information modeling are used as the input of the ON-LSTM structure, which can better capture the continuity of emotion changes in time for the emotion recognition of the EEG signal with the 4D structure, is of great significance.

### 3.5. Method Comparison

In order to show that the 4D-FSTception method is superior to other classification methods, this article chooses the following EEG emotion classification methods for comparison.

HCNN [38]: It uses a hierarchical CNN for EEG emotion classification and recognition and uses the differential entropy features of two-dimensional EEG as the input of the neural network model, which proves that the band and band are more suitable for emotion recognition. The method considers the spatial information and frequency information of the EEG signal.RGNN [39]: It uses the adjacency matrix in the graph neural network to simulate the inter-channel relationship in the EEG signal and realizes the simultaneous capture of the relationship between the local channel and the global channel. The connection and sparseness of the adjacency matrix are determined by humans. Supported by the neurological theory of brain tissue, this method shows that the relationship between global channels and local channels in the left and right hemispheres plays an important role in emotion recognition.BDGLS [40]: It uses differential entropy features as input data. By combining the advantages of dynamic graph convolutional neural networks and generalized learning systems, emotion recognition accuracy can be improved over the full frequency band of EEG features. This method considers the frequency information and spatial information of the EEG signal at the same time.PCRNN [32]: It first uses the CNN module to obtain space characteristics from each 2D EEG topographic map, then uses LSTM to obtain time characteristics from the EEG vector sequence, and finally integrates space and time characteristics to carry out emotional classification.4D-CRNN [41]: It first extracts features from EEG signals to construct a four-dimensional feature structure, then uses convolutional recurrent neural networks to extract EEG signals to obtain spatial features and frequency features, and uses LSTM to extract time from EEG vector sequences features, and finally carry out EEG emotion classification.4D-aNN (DE) [42]: It uses 4D space-spectrum-time representations containing the space, frequency spectrum, and time information of the EEG signal as input. An attention mechanism is added to the CNN module and the bidirectional LSTM module. This method also considers the time, space, and frequency of EEG information.ACRNN [43]: It adopts the convolutional recurrent neural network method based on the attention mechanism. It first uses the attention mechanism to distribute the weights between channels, then uses the CNN to extract the spatial information of the EEG, and finally uses the RNN to integrate and extract the temporal information features of the EEG. The method considers spatial information and temporal information for the emotion classification of EEG signals.

The above methods considered in the statistical validation experiments in this paper mainly include HCNN, RGNN, and BDGLS methods that used only frequency and spatial information for sentiment classification and recognition, as well as PCRNN and ACRNN methods that used spatial and temporal information features for sentiment classification. We also worked with 4D-CRNN and 4D-aNN methods that used frequency, spatial, and temporal features for sentiment recognition at the same time.

We reproduce these methods described above based on the structural parameters exploited in the original paper. We conduct five-fold cross-validation experiments for these approaches on the SEED and DEAP data. We list the mean accuracy and standard deviation for each method on the SEED data set and DEAP data set in Table 10. For the SEED data set, the ACC of 4D-FSTception is 95.49%, which is 6.89% higher than the HCNN method that only considers the spatial information of the EEG signal, and it is also 1.25% higher than the RGNN. For the frequency information of the EEG signal simultaneously and the BDGLS method of spatial information, this method is 1.83% higher than this method. Both the 4D-CRNN and 4D-aNN (DE) methods consider the frequency, space, and time features at the same time. We choose the same input feature structure, and the differential entropy feature is selected in the frequency feature extraction. The 4D-FSTception is 0.75% higher than 4D-CRNN and 0.6% higher than 4D-aNN. For the DEAP data set, the experimental method in this paper achieves good results in EEG emotion recognition; its valence and arousal are 94.61% and 95.02%, respectively, which are 0.39% and 0.44% higher than 4D-CRNN, respectively. It is 4.35% and 4.04% higher than PCRNN, which only considers frequency and temporal information, and 0.89% and 1.64% higher than ACRNN, which only considers spatial and temporal information. For the 4D-CRNN method that simultaneously considers frequency, spatial, and temporal features, our method is 0.39% and 0.44% higher than 4D-CRNN in valence and arousal, respectively.

To sum up, the method proposed in this paper shows the best emotion recognition and classification performance in any combination of frequency, time, and space information or in the combination of the three characteristics.

## 4. Conclusions

The paper points out a 4D-FSTception method for EEG emotion identification. The 4D input features include the frequency features of different frequency bands of the EEG signal, as well as the continuous-time features, while also maintaining the spatial information between channels. This method first extracts the differential entropy feature vectors of different frequency bands of the EEG signal at the same time. Secondly, according to the distribution position of the electrodes, the multichannel one-dimensional feature vector is converted into a two-dimensional feature matrix, which fully considers the spatial position information of the EEG channel. Then, the two-dimensional feature matrix is superimposed to form a three-dimensional matrix, and finally, the three-dimensional EEG sequence time segments divided into fixed lengths constitute 4D input features. Next, we introduce the FSTception model closely combined with CNN and ON-LSTM in detail. CNN is used to solve frequency and space data, and ON-LSTM is used to obtain time correlations from CNN output to reveal the underlying deep emotional characteristics. The proposed method achieves better performance on both SEED and DEAP datasets. By comparing with seven excellent studies, the 4D-FSTception model can effectively improve the accuracy of emotion discrimination in EEG signals. We hope to enhance our paper by building our own data set and verifying the validity of our approach in a real-world scenario. In future work, we can consider combining DE features with more features to further enhance the emotion recognition performance of EEG signals through feature complementarity.

## Figures and Tables

**Figure 1 entropy-24-01830-f001:**
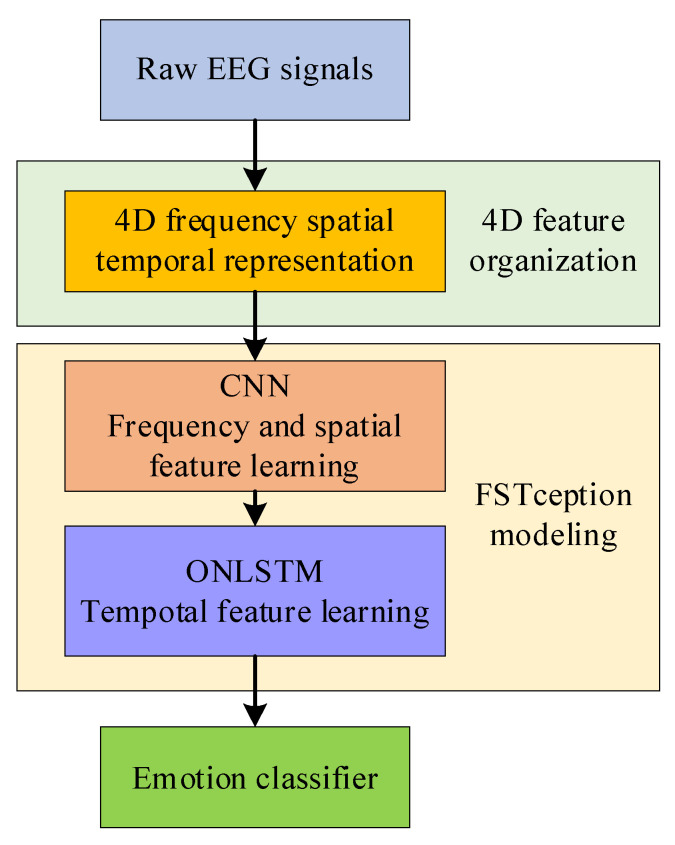
The general flowchart of the proposed method.

**Figure 2 entropy-24-01830-f002:**
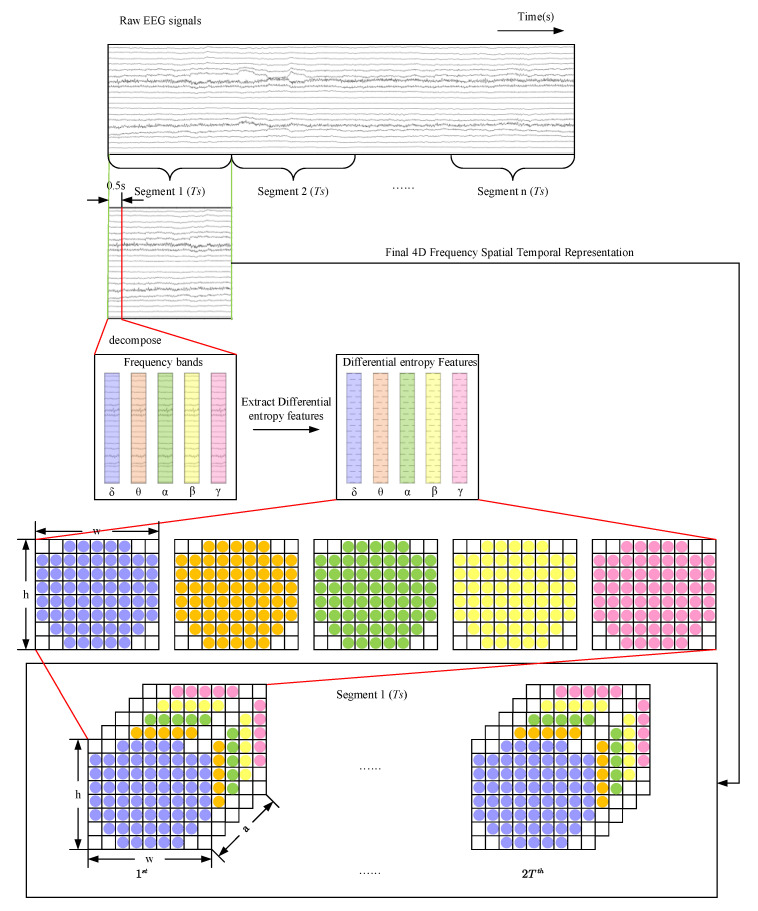
The generation of 4D inputs.

**Figure 3 entropy-24-01830-f003:**
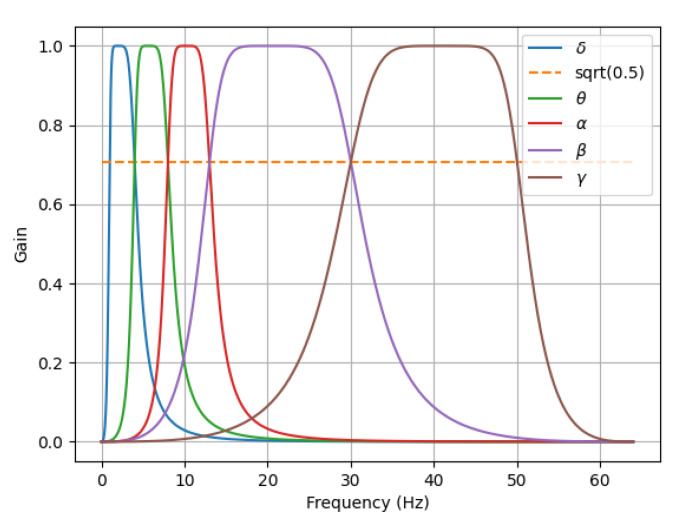
The frequency response of the filter.

**Figure 4 entropy-24-01830-f004:**
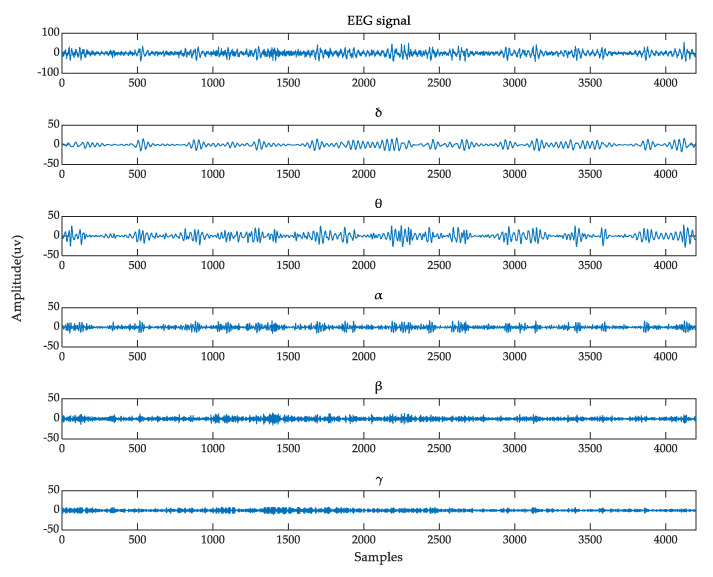
The raw EEG signal and its five frequency bands in the DEAP dataset.

**Figure 5 entropy-24-01830-f005:**
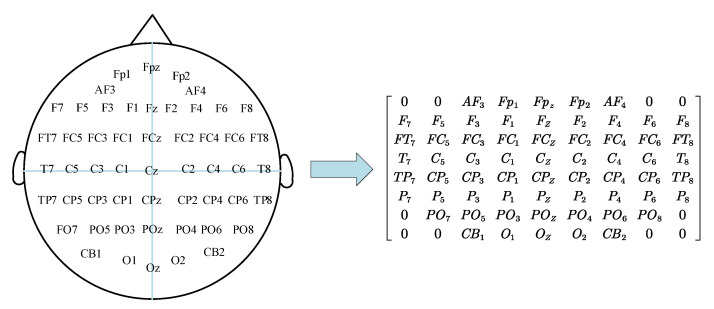
The 2D map of 62 channels.

**Figure 6 entropy-24-01830-f006:**
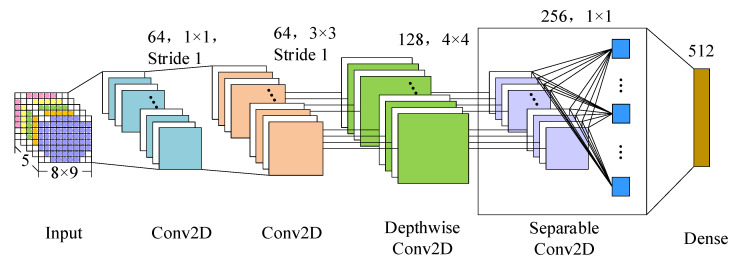
The structure of CNN module for frequency and spatial feature learning.

**Figure 7 entropy-24-01830-f007:**
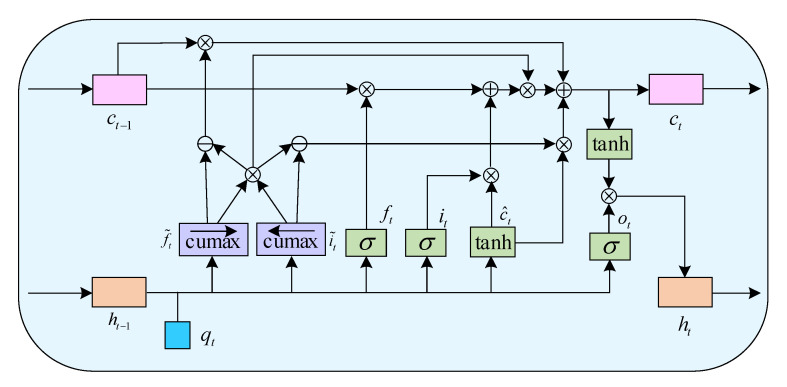
The detailed structure of ON-LSTM.

**Figure 8 entropy-24-01830-f008:**
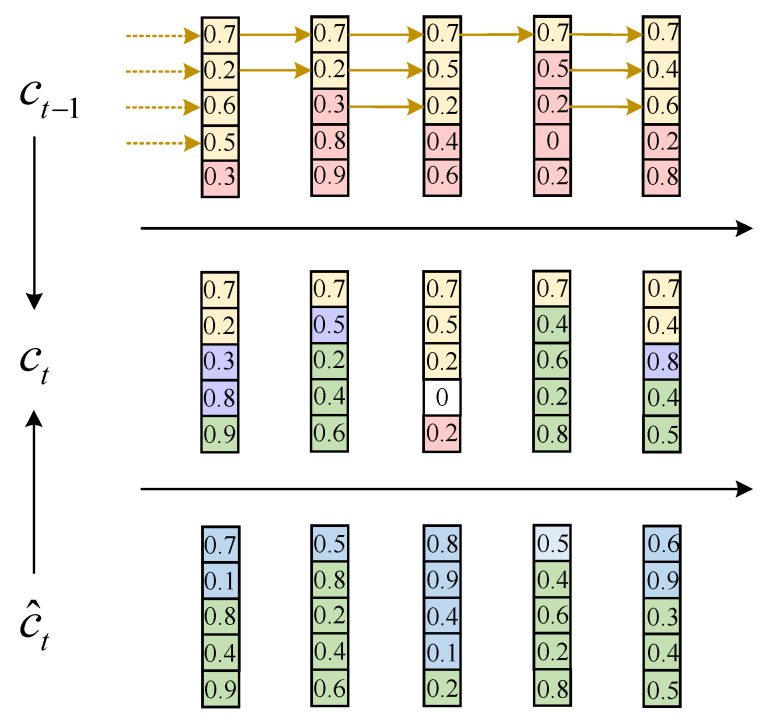
ON-LSTM partition update. The numbers are randomly generated, with historical information at the top, current input at the bottom, and currently integrated output in the middle. The yellow part at the top is the historical information level (the main forgetting gate), the green part at the bottom is the input information level (the main input gate), the yellow part in the middle is the directly copied historical information, the green is the directly copied input information, the purple is the intersection information fused according to the LSTM, and the white is the unrelated “blank area”.

**Figure 9 entropy-24-01830-f009:**
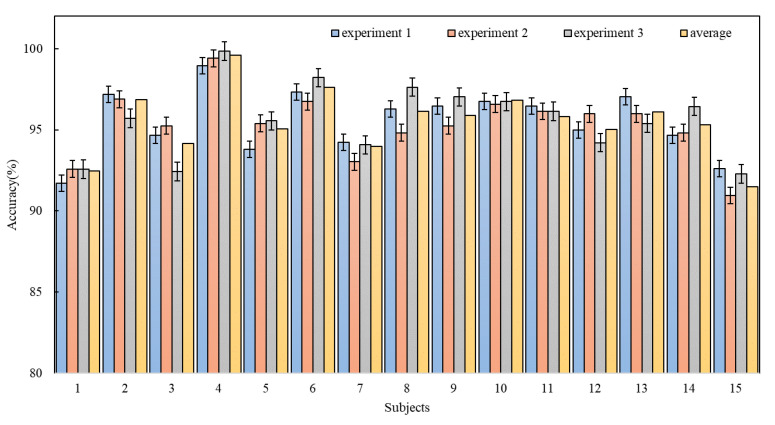
Overall performance of the 4D-FSTception model on SEED dataset.

**Figure 10 entropy-24-01830-f010:**
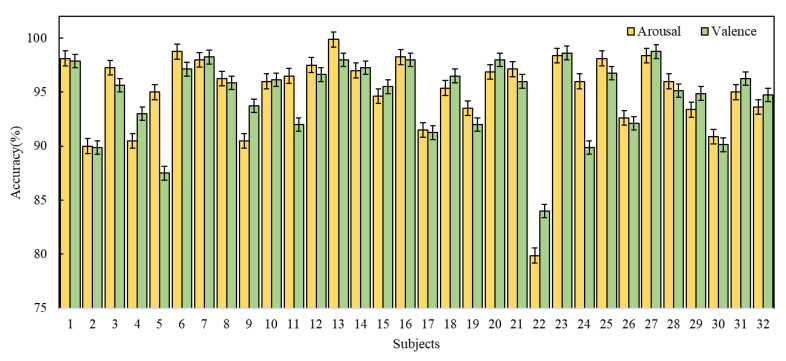
Overall performance of the 4D-FSTception model on DEAP dataset.

**Figure 11 entropy-24-01830-f011:**
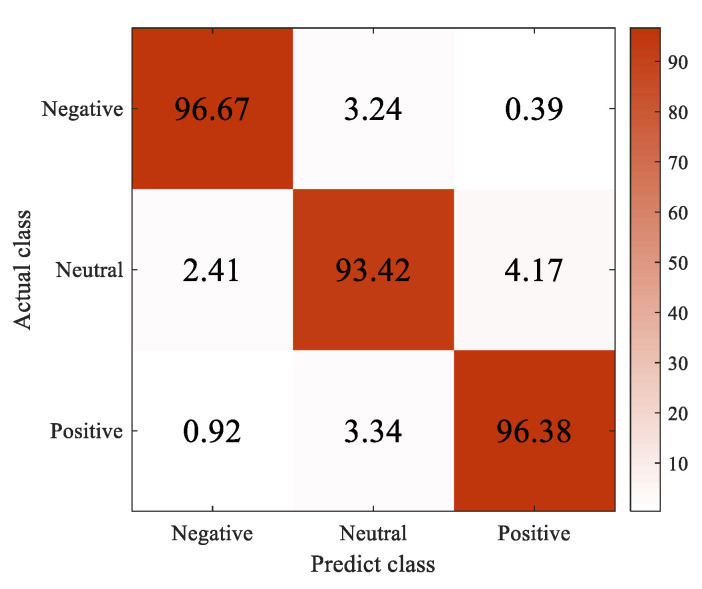
Confusion matrices of our method on SEED dataset.

**Figure 12 entropy-24-01830-f012:**
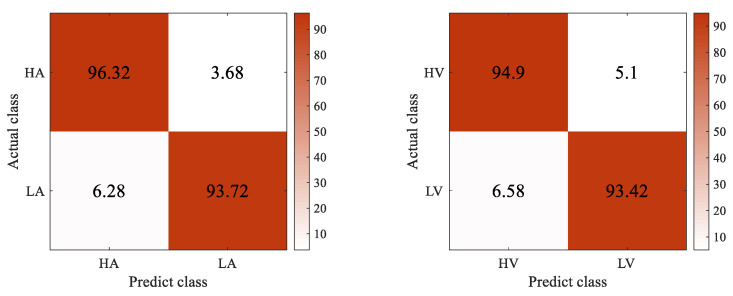
Confusion matrix: (**left**) arousal, (**right**) valence on DEAP dataset.

**Figure 13 entropy-24-01830-f013:**
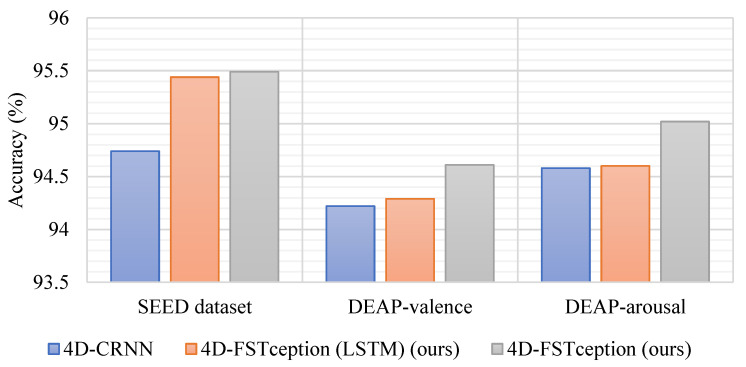
Accuracy performance of different methods in different datasets.

**Figure 14 entropy-24-01830-f014:**
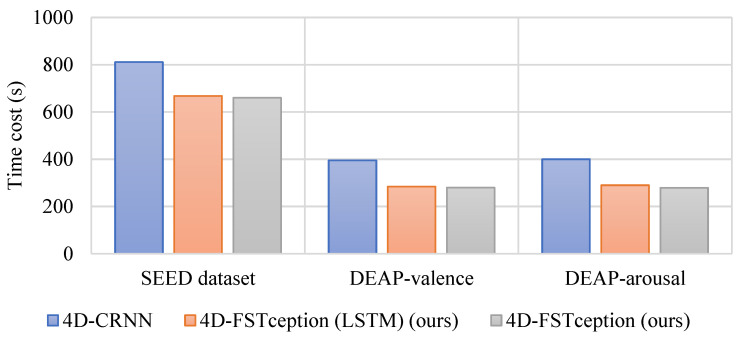
Time cost performance of different methods in different datasets.

**Figure 15 entropy-24-01830-f015:**
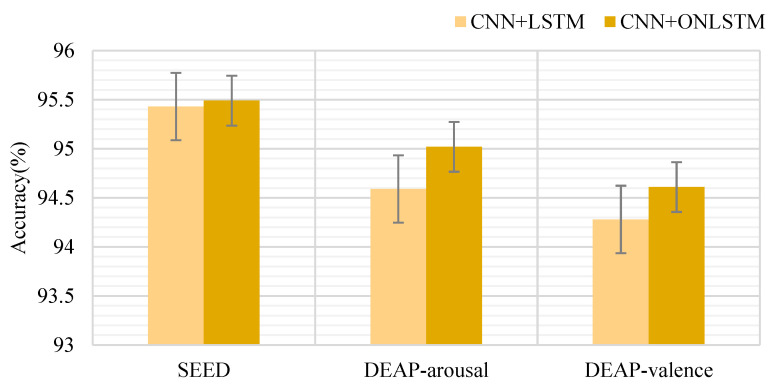
CNN + LSTM and CNN + ON-LSTM compare the experimental results.

**Table 1 entropy-24-01830-t001:** The detailed comparison to other works.

Article	Method	Application	Dataset	Accuracy (%)
Komolovaitė et al. [23]	EEGNet SSVEP	Alzheimer research	Figshare website	50.2
Thammasan et al. [24]	Deep Belief Network (DBN)	Emotion recognition in music listening	15 recruited healthy students (valence/arousal emotion types)	82.42/88.24
Tripathi et al. [25]	Convolutional Neural Network (CNN)	Automatic emotion recognition	DEAP (valence/arousal emotion types)	73.36/81.41
Salama et al. [26]	3-Dimensional Convolutional Neural Networks (3D-CNN)	Emotion recognition	DEAP (valence/arousal emotion types)	88.49/87.44
Yang et al. [27]	Convolutional Neural Network (CNN)	Automatic emotion recognition	DEAP (valence/arousal emotion types)	90.24/89.45
Zheng et al. [29]	Deep Belief Network (DBN)	Emotion recognition	15 recruited subjects	86.65
Meng-meng et al. [30]	Common Spatial Pattern (CSP)	Emotion recognition	6 recruited healthy students	87.54
Liu et al. [31]	Recurrent Convolutional Neural Network and Long Short Term Memory (RCNN-LSTM)	Automatic emotion recognition	DEAP	96.63

**Table 2 entropy-24-01830-t002:** The frequency modes and corresponding characteristics of EEG.

Patterns	Frequency	Brain State	Awareness
(δ) Delta	1–4 Hz	Deep sleep pattern	Lower
(θ) Theta	4–8 Hz	Light sleep pattern	Low
(α) Alpha	8–13 Hz	Closing the eyes, relax state	Medium
(β) Beta	13–30 Hz	Active thinking, focus, high alert, anxious	High
(γ) Gamma	30–50 Hz	Mentally active and hypertensive	Higher

**Table 3 entropy-24-01830-t003:** DEAP dataset content.

Name	Size	Contents
Data	40 × 40 × 8064	videos × channels × data
Labels	40 × 4	videos × labels (valence, arousal, dominance, liking)

**Table 4 entropy-24-01830-t004:** Recognition Accuracy for each subject on SEED dataset.

Subjects	Accuracy	Subjects	Accuracy	Subjects	Accuracy	Subjects	Accuracy
1	92.45%	5	95.06%	9	95.89%	13	96.12%
2	96.86%	6	97.63%	10	96.83%	14	95.32%
3	94.17%	7	93.99%	11	95.83%	15	91.50%
4	99.59%	8	96.15%	12	95.03%		

**Table 5 entropy-24-01830-t005:** Recognition Accuracy for each subject on “Arousal”.

Subjects	Accuracy	Subjects	Accuracy	Subjects	Accuracy	Subjects	Accuracy
1	98.13%	9	90.50%	17	91.50%	25	98.13%
2	90.00%	10	96.00%	18	95.38%	26	92.63%
3	97.25%	11	96.50%	19	93.50%	27	98.38%
4	90.50%	12	97.50%	20	96.88%	28	96.00%
5	95.00%	13	99.86%	21	97.13%	29	93.38%
6	98.75%	14	97.00%	22	79.88%	30	90.88%
7	98.00%	15	94.63%	23	98.38%	31	95.00%
8	96.25%	16	98.25%	24	96.00%	32	93.63%

**Table 6 entropy-24-01830-t006:** Recognition Accuracy for each subject on “Valence”.

Subjects	Accuracy	Subjects	Accuracy	Subjects	Accuracy	Subjects	Accuracy
1	97.88%	9	93.75%	17	91.25%	25	96.75%
2	89.88%	10	96.13%	18	96.50%	26	92.13%
3	95.63%	11	92.00%	19	92.00%	27	98.75%
4	93.00%	12	96.63%	20	98.00%	28	95.13%
5	87.50%	13	98.00%	21	96.00%	29	94.88%
6	97.13%	14	97.25%	22	84.00%	30	90.13%
7	98.25%	15	95.50%	23	98.63%	31	96.25%
8	95.88%	16	98.00%	24	89.88%	32	94.75%

**Table 7 entropy-24-01830-t007:** The score of 4D-FSTception evaluation index is on SEED dataset.

Classes	Accuracy	Precision	Recall	F1-Score
Negative	96.67%	94.87%	92.50%	93.67%
Neutral	93.42%	94.29%	93.94%	96.12%
Positive	96.38%	98.96%	95.00%	96.94%

**Table 8 entropy-24-01830-t008:** The score of 4D-FSTception evaluation index is on DEAP dataset.

Valence/Arousal	Class	Accuracy	Precision	Recall	F1-Score
Valence	High	94.90%	98.48%	93.94%	96.47%
	Low	93.42%	98.41%	93.18%	96.12%
Arousal	High	96.32%	98.02%	91.67%	94.74%
	Low	93.72%	93.30%	95.37%	96.71%

**Table 9 entropy-24-01830-t009:** Comparison of computational performance with other methods.

Method	Map Shape	SEED		DEAP-Valence		DEAP-Arousal			
		Acc (%)	Time Cost (s)	Acc (%)	Time Cost (s)	Acc (%)	Time Cost (s)	FLOPS (G)	Params (M)
HCNN [38]	19 × 19	88.60	3600	-	-	-	-	-	-
4D-CRNN [42]	8 × 9	94.74	811	94.22	395	94.58	400	18.63	15.8
4D-FSTception (LSTM) (ours)	8 × 9	95.44	668	94.29	284	94.60	290	19.94	14.4
4D-FSTception (ours)	8 × 9	95.49	660	94.61	280	95.02	279	20.61	12.7

**Table 10 entropy-24-01830-t010:** The performances (average ACC ± STD (%)) of the compared methods on SEED and DEAP datasets.

Method	Information	ACC ± STD (%)		
		SEED	DEAP-Valence	DEAP-Arousal
HCNN [38]	Frequency + spatial	88.60 ± 2.60	-	-
RGNN [39]	Frequency + spatial	94.24 ± 5.95	-	-
BDGLS [40]	Frequency + spatial	93.66 ± 6.11	-	-
PCRNN [32]	Spatial + temporal	-	90.26 ± 2.88	90.98 ± 3.09
ACRNN [43]	Spatial + temporal	93.72	3.21	93.38
4D-CRNN [41]	Frequency + spatial + temporal	94.74 ± 2.32	94.22 ± 2.61	94.58 ± 3.69
4D-aNN (DE) [42]	Frequency + spatial + temporal	95.39 ± 3.05	-	-
4D-FSTception (LSTM) (ours)	Frequency + spatial + temporal	95.44 ± 0.32	94.29 ± 1.89	94.60 ± 2.08
4D-FSTception (ours)	Frequency + spatial + temporal	95.49 ± 3.01	94.61 ± 2.83	95.02 ± 2.85

## Data Availability

Datasets are available in a publicly accessible repository that does not issue DOIs. These data can be found at the following address: http://www.eecs.qmul.ac.uk/mmv/datasets/deap/index.html (accessed on 20 July 2022), and https://bcmi.sjtu.edu.cn/home/seed/(accessed on 16 March 2021).

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
