# Peer review of "Multidimensional Feature in Emotion Recognition Based on Multi-Channel EEG Signals"

_entropy, 2022, doi:10.3390/e24121830_

Round 1

Reviewer 1 Report

The authors present a new approach to emotion recognition based on EEG signal analysis. Although the authors present a potentially useful method the paper requires significant improvements.

Very annoying is the language surface. The paper starts with the repetition "cognitive science" in the first sentence. Many punctuation and grammar mistakes which I am not going to list here.

The more serious problem is with the methods description and experiment presentation/design.

1. variance definition eq. 3 is valid only is \mu = 0, but it is not stated

2. It is not clear that the variance of electrical signal (potential?) is equal to its energy - required explanation

3. line 178 How "i" could be "the same"? 

4. How usage of DE can reduce the error (uncertainty?)? Higher quality of a measurement is achieved by increasing the number of observations or quality of equipment. Not by data transformation.

 5. What does it mean "N" l.172 - normal distribution, l 173 - number of observations. The notations should be clarified.

6. How many data points are used to estimate ED? The length of the time interval was 0.5s, frequency 1-4 Hz, so 0.5 up to 2 data points was used to estimate DE. It is too small for continuous variables. Probably some explanation is missing. However, even in the optimistic assumption that the probing frequency was 128 Hz, that the single chunk was 74 data points - not much considering the necessity of probability distribution estimation. The situation could be improved by assuming that it was a normal distribution, but then any normality test should be applied to justify the distribution choice.

I suggest removing lines from figures 5-8. They have no meaning but generate a complete mess on the graphs. I would consider also replacing it with a table. 

Assuming that the authors would answer the raised questions the paper might be considered for publication.

Reviewer 2 Report

In this work, a methodology to detect emotions from EEG signals is presented. In general, the work is interesting and well-written. Also, a comparison with other reported works is presented. Although promising results are obtained, some minor issues have to be addressed for clarity.

Please describe all the acronyms, EEG, DEAP, SEED, LSTM, etc.

How do you determine which frequency, spatial and temporal characteristics are important, i.e., they can be redundant or not provide significant information?

Line 57: Why is a Butterworth filter used? There are better filter techniques. Please provide its frequency response.

Provide a general flowchart about your proposal (lines 121-131).

Line 197: Which is the impact of using other values? The same happens for the values in lines 210-212, line: 359, etc.. Please justify all your decisions.

Which is the meaning of colors in Figure 5?

For readers, include some EEG signals.

EEG data preprocessing is fundamental for the achievement of your results, but any results are provided. Please show in a detailed way the steps and their respective results described in section 3.2.

It is not clear which are the DE features shown in Figure 1 since no results about them are presented. Please describe which features are used and how they are analyzed/selected in order to avoid information redundancy.

There is an error in table 3; also, discuss computational time.

Please provide the confusion matrices and their metrics, accuracy, precision, recall, and f1-score.

Table 2: Provide a graph to observe and compare maxima and minima values in an easier way.

In figures 5-7, please discuss possible reasons to obtain different results for each subject, mainly the one that provides the worst performance.

Finally, describe the limitations of your work and future work on the topic. 

Reviewer 3 Report

The author proposes a new approach considering the 4D structure of EEG signal processing for emotion identification. The authors make an effort to build the model and compare it with previous models. It is a fair comparison of DEAP and SEED and shows improvement using the proposed method. The presentation is clear.   I think the model trained each participant. Did you confirm sufficient data size from a single subject to train the model? It might be applicable leave-one-subject-out cross-validation to validate the generalizability of the models for future work.   I recommend providing statistical testing comparing the proposed model and the previous model.   Please carefully revise the manuscript, for it includes some abbreviations without definitions and inappropriate hyphens.

Round 2

Reviewer 2 Report

All the comments and suggestions have been adequately addressed. This reviewer recommends the manuscript's acceptance. 

Author Response

We thank you for your response and your decision to suggest that the manuscript be accepted.